# 4-alkyl-L-(Dehydro)proline biosynthesis in actinobacteria involves *N*-terminal nucleophile-hydrolase activity of γ-glutamyltranspeptidase homolog for C-C bond cleavage

Guannan Zhong[1], Qunfei Zhao[1,2], Qinglin Zhang[3] & Wen Liu[1,2,3]

γ-Glutamyltranspeptidases (γ-GTs), ubiquitous in glutathione metabolism for γ-glutamyl transfer/hydrolysis, are *N*-terminal nucleophile (Ntn)-hydrolase fold proteins that share an autoproteolytic process for self-activation. γ-GT homologues are widely present in Gram-positive actinobacteria where their Ntn-hydrolase activities, however, are not involved in glutathione metabolism. Herein, we demonstrate that the formation of 4-Alkyl-L-(dehydro)proline (ALDP) residues, the non-proteinogenic α-amino acids that serve as vital components of many bioactive metabolites found in actinobacteria, involves unprecedented Ntn-hydrolase activity of γ-GT homologue for C–C bond cleavage. The related enzymes share a key Thr residue, which acts as an internal nucleophile for protein hydrolysis and then as a newly released *N*-terminal nucleophile for carboxylate side-chain processing likely through the generation of an oxalyl-Thr enzyme intermediate. These findings provide mechanistic insights into the biosynthesis of various ALDP residues/associated natural products, highlight the versatile functions of Ntn-hydrolase fold proteins, and particularly generate interest in thus far less-appreciated γ-GT homologues in actinobacteria.

[1] State Key Laboratory of Bioorganic and Natural Products Chemistry, Shanghai Institute of Organic Chemistry, Chinese Academy of Sciences, 345 Lingling Road, Shanghai 200032, China. [2] State Key Laboratory of Microbial Metabolism, School of Life Science & Biotechnology, Shanghai Jiao Tong University, 800 Dongchuan Road, Shanghai 200240, China. [3] Huzhou Center of Bio-Synthetic Innovation, 1366 Hongfeng Road, Huzhou 313000, China. Correspondence and requests for materials should be addressed to W.L. (email: wliu@mail.sioc.ac.cn).

γ-Glutamyltranspeptidases (γ-GTs), which catalyse the transfer/hydrolysis of γ-glutamyl from the thiol glutathione (GSH), GSH S-conjugates or glutamine, are ubiquitous enzymes in living organisms that play critical roles during antioxidant defence, detoxification and inflammatory processes[1]. These enzymes belong to the superfamily of N-terminal nucleophile (Ntn)-hydrolase fold proteins, and exhibit the activity for post-translational autocatalytic cleavage of a peptide precursor to form a functional heterodimer in which a newly released N-terminal residue acts as a nucleophile and mediates amide bond hydrolysis through the formation of an acyl-nucleophile enzyme intermediate[2]. Intriguingly, γ-GT homologues have been found to be widely present in the organisms that do not involve GSH metabolism, such as Gram-positive actinobacteria[3–5]. The results from their Ntn-hydrolase activities in these bacteria, many of which in fact are important antibiotic-producing actinomycetes, have not been well appreciated to-date.

4-Alkyl-L-(dehydro)proline (ALDP) residues, which differ in their oxidation states and extents of carbon side-chain functionalization, constitute a class of α-amino acids that are non-proteinogenic but serve as vital components of a number of active small molecules produced by actinomycetes (Fig. 1a). These metabolites, which include pyrrolobenzodiazepines (PBDs), hormaomycins and lincomycin A, possess distinct architectures that result in a wide variety of biological properties. Many PBDs, such as anthramycin, porothramycin, sibiromycin and tomaymycin, in which an ALDP residue conjugates to an anthranilate derivative to form a seven-membered 1,4-diazepine-pharmaceutical ring system, are sequence-specific DNA-alkylating agents that display remarkable activities against tumour cells[6]. Hormaomycins are a group of highly modified ALDP-containing

cyclodepsipeptides that act as microbial hormones to stimulate the production of antibiotics and the formation of aerial mycelium in various actinomycetes[7–9]. In contrast, lincomycin A, which features an eight-carbon aminosugar central to an N-methylated ALDP residue (that is, trans-4-propyl-L-proline, PPL) and a methylmercapto group, is an antibiotic that has been widely used for half a century to treat Gram-positive bacterial infections[10–13].

Biosynthetically, different ALDPs originate primarily from a common proteinogenic α-amino acid, L-tyrosine, which undergoes an oxidation-associated ring-opening and re-closure process for the rearrangement of its benzoic carbon skeleton to a pyrroline moiety that contains a carboxylate side chain (Fig. 1b)[14–16]. Specifically, an L-tyrosine hydroxylase (for example, LmbB2 in lincomycin biosynthesis) catalyses an ortho-hydroxylation reaction to produce L-3,4-dihydroxy-phenylalanine (L-DOPA, 1); subsequently, a dioxygenase (for example, LmbB1) acts on 1 to perform extradiol cleavage, and the cyclization of the resulting semialdehyde intermediate yields an unstable pyrroline product in the form of either an imine (2) or its tautomer enamine (3)[17–20]. The following steps remain poorly understood; however, removing the terminal two-carbon (2-C) unit from the side chain of 2 or 3 appears to be necessary[16]. This tailoring would generate the next shared intermediate, diene 4, which could undergo variable modifications towards the biosynthesis of different ALDP members (Fig. 1b). Focusing on this key reaction involved in carbon chain processing, herein, we utilize Streptomyces lincolnensis, the producer of lincomycin A, as a model system to facilitate our search for the corresponding enzymatic activity. Ultimately, this effort revealed for the first time that LmbA-like γ-GT homologues exhibit unusual Ntn-hydrolase activity to catalyse C–C bond cleavage, despite all

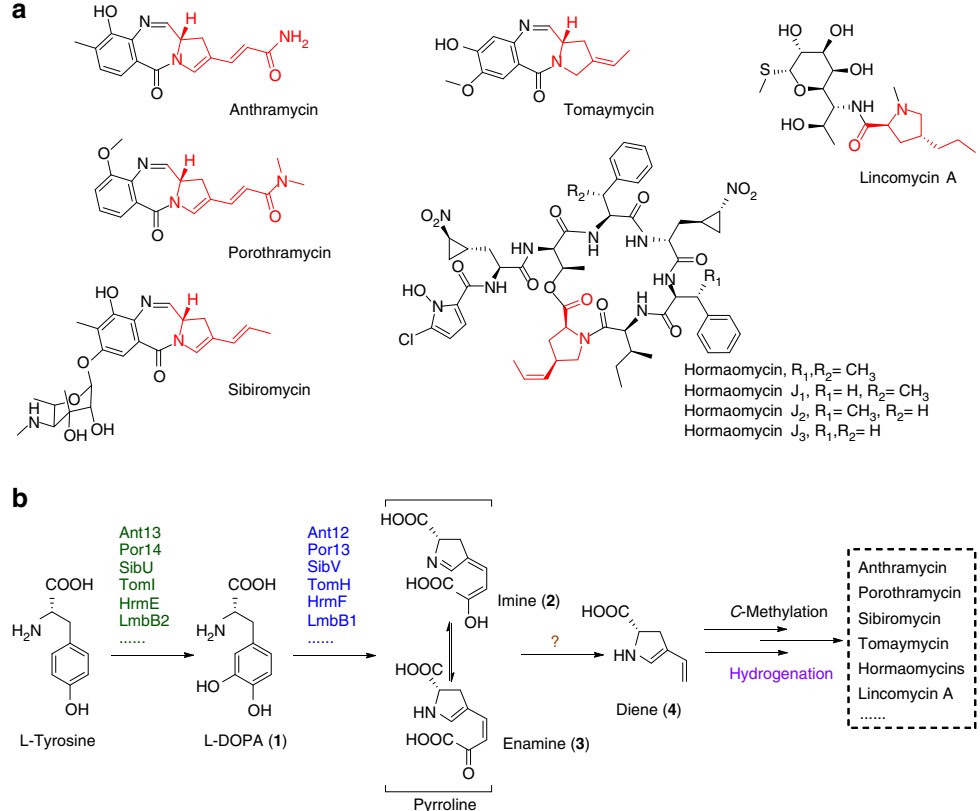

**Figure 1 | Chemical structures and biosynthetic pathway.** (**a**) Representative ALDP (highlighted in red)-containing natural products. (**b**) Common biosynthetic steps (differentiated in colour) during the formation of various ALDP residues.

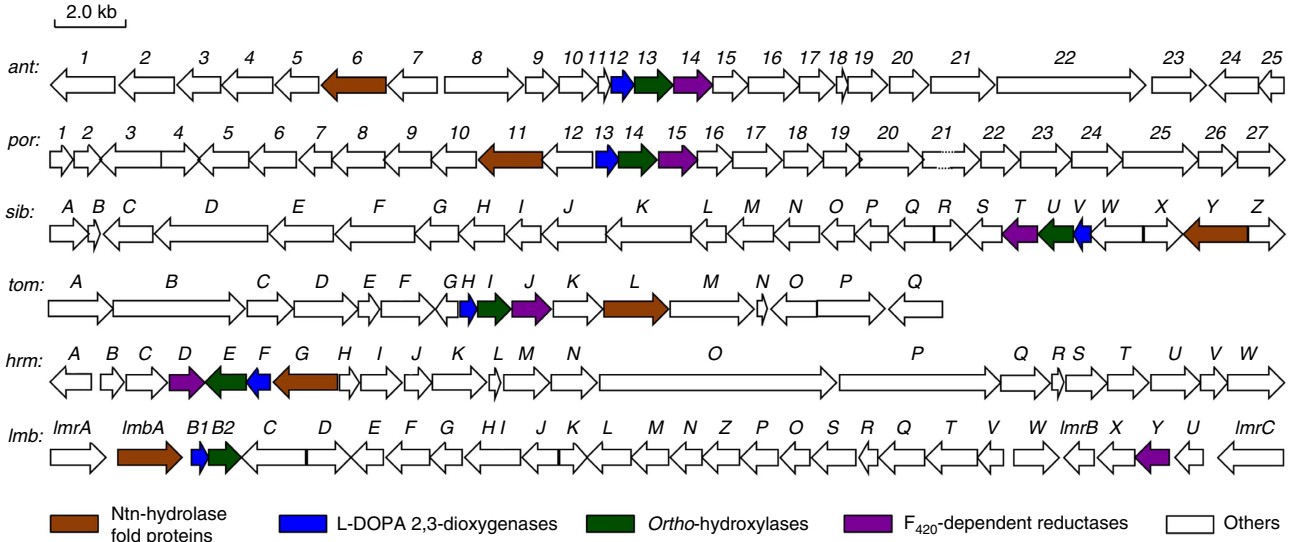

**Figure 2 | Comparative analysis of available ALDP-related biosynthetic gene clusters.** The genes coding for ALDP formation are coloured. *ant*, for anthramycin; *por*, for porothramycin; *sib*, for sibiromycin; *tom*, for tomaymycin; *hrm*, for hormaomycins and *lmb* for lincomycin A. For sequence identities of deduced proteins, LmbA is homologous to Ant6 (72.3%), Por11 (72.9%), SibY (58.0%), TomL (50.0%) and HrmG (64.6%); LmbY is homologous to Ant14 (49.8%), Por15 (48.2%), SibT (53.5%), TomJ (50.2%) and HrmD (46.3%); LmbB1 is homologous to Ant12 (40.8%), Por13 (44.6%), SibV (54.4%), TomH (52.8%) and HrmF (46.2%); and LmbB2 is homologous to Ant13 (31.2%), Por14 (30.9%), SibU (38.7%), TomI (37.3%) and HrmE (42.9%).

members of this superfamily characterized to-date exhibiting only amidohydrolase activity for C–N bond cleavage.

## Results

**Assignment of *lmbA* and its homologues in ALDP formation.** Comparative analysis of the available biosynthetic gene clusters of ALDP-containing metabolites[21–27], that is, lincomycin A (*lmb*), anthramycin (*ant*), porothramycin (*por*), sibiromycin (*sib*), tomaymycin (*tom*) and hormaomycins (*hrm*), revealed a set of highly conserved genes, for example, *lmbB1*, *lmbB2*, *lmbY* and *lmbA* in *lmb* cluster, from which the deduced proteins are thus readily correlated with the common reactions occurring during the biosynthesis of ALDP residues (Fig. 2). The oxidase pairs, for example, LmbB1 and LmbB2, are known to be involved in the transformation of L-tyrosine into pyrroline intermediate **2** or **3** (refs 17–20), whereas the roles of LmbY- and LmbA-like proteins remain unclear.

LmbY and its counterparts (that is, Ant14, Por15, SibT, TomJ and HrmD), which share sequence homology with putative coenzyme $F_{420}$-dependent oxidoreductases, are believed to function in olefinic double bond reduction at a later stage of ALDP formation[16,28]. The biosynthesis of the $F_{420}$ cofactor involves glutamyl transfer activity[29]. Despite the lack of experimental evidence, this activity has long been speculated to arise from LmbA and its counterparts (that is, Ant6, Por11, SibY, TomL and HrmG) because they are homologous to various γ-GTs. γ-GTs catalyse γ-glutamyl transfer/hydrolysis with Ntn-hydrolase activity through an acyl-nucleophile enzyme intermediate[2]. Mechanistically, the production of **4** in the biosynthetic pathways of ALDP residues could occur through the hydrolysis of **3** to release the diene group (Fig. 1b). This process would require nucleophilic attack by water onto the α-keto group of the side chain to produce the co-product, oxalic acid, either directly or indirectly after the formation of an acyl-nucleophile enzyme intermediate. Therefore, in the search for an alternative to the cofactor-forming proteins for supporting LmbY-like activity, the mechanistic consistency inspired us to ask whether LmbA and its counterparts share a similar acyl

transfer/hydrolysis process with γ-GTs during the conversion of pyrroline intermediate **3** into diene product **4**. However, no evidence exists suggesting that the known γ-GTs and other Ntn-hydrolase fold proteins possess C–C bond cleavage activity.

**Correlation of related genes with ALDP formation.** To validate the necessity of LmbA-like activity for ALDP formation, we inactivated *lmbA* in the lincomycin-producing *S. lincolnensis* strain (Supplementary Fig. 1). This mutation substantially lowered the yield of lincomycin A by ∼60-fold relative to that of the wild-type strain; then, the *in trans* expression of *lmbA* in the Δ*lmbA* mutant strain partially restored the production capacity, producing lincomycin A in a yield of ∼20% of that produced by the wild-type strain (Fig. 3a). Clearly, LmbA plays a vital role in lincomycin biosynthesis. A similar complementation effect was found by substituting *lmbA* with *ant6*, the *lmbA* homologue in *ant* cluster (Fig. 3a), demonstrating the functional identity between LmbA and its pathway-specific counterparts for ALDP formation.

The Δ*lmbA* mutant strain still produced lincomycin A (Fig. 3a), indicating that an orthologue outside *lmb* cluster partially compensates for the loss of *lmbA*. Indeed, sequence analysis of the *S. lincolnensis* genome revealed *lmbA2991* (48.4% identity to *lmbA*) (Supplementary Fig. 2). As anticipated, further inactivation of this gene in the Δ*lmbA* mutant strain led to the complete abolishment of lincomycin production (Fig. 3a). Importantly, either the single (Δ*lmbA*) or the double (Δ*lmbA-lmbA2991*) mutant strain accumulated intermediate **6**, the thiol ergothioneine (EGT) *S*-conjugated lincosamide derivative, but did not produce the specific ALDP residue PPL; in contrast, the Δ*lmbD* mutant control strain produced both (Fig. 3a)[30]. We recently established that a discrete non-ribosomal peptide synthetase system involving LmbD activity catalyses the assembly of **6** and PPL to provide a key EGT *S*-conjugated intermediate (**7**) in the lincomycin biosynthetic pathway (Fig. 3b)[30]. Clearly, the significant decrease (for the Δ*lmbA* single mutation) or complete abolishment (for the Δ*lmbA-lmbA2991* double mutation) of lincomycin production could be attributed to the lack of LmbA or both LmbA and

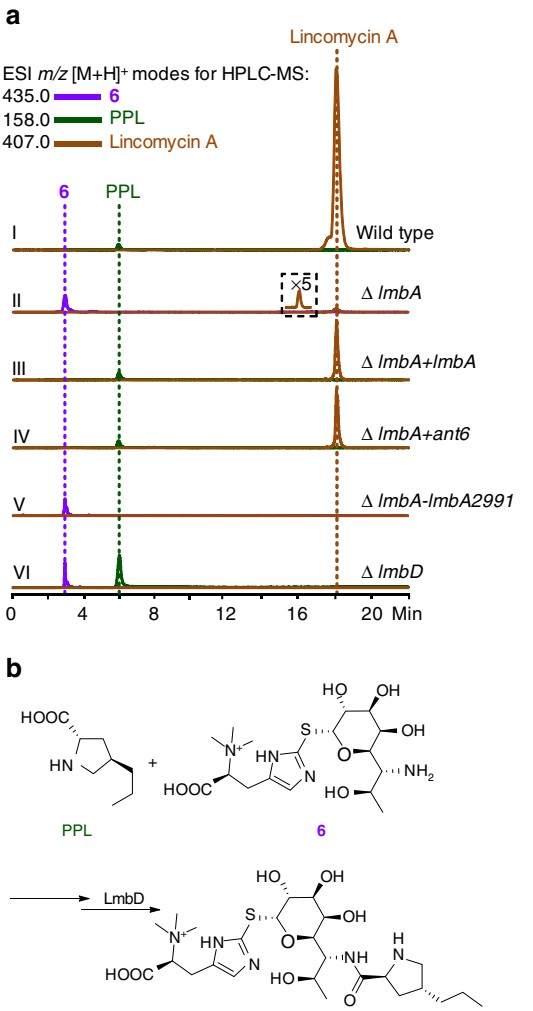

**Figure 3 | In vivo functional characterization of lmbA and its orthologue lmbA2991.** (**a**) Product profiles of *S. lincolnensis* strains, including the wild-type control (I) and mutants (II, for the ΔlmbA single mutation; III, for homologous complementation of the ΔlmbA mutant by lmbA; IV, for heterologous complementation of the ΔlmbA mutant by ant6; V, for ΔlmbA-lmbA2991 double mutation; VI, for the ΔlmbD single mutation). (**b**) The assembly of PPL and **6** that is mediated by a discrete NRPS system involving LmbD activity.

LmbA2991 activities in the mutant strain, which caused PPL supply and, thus, its subsequent assembly with **6** for molecular tailoring and maturation, to fail.

**Typical autoproteolytic activity for self-activation**. Despite numerous attempts, the expression of LmbA and its counterparts (for example, Ant6 and SibY) remained highly refractory to various approaches for soluble protein preparation in *Escherichia coli*. Because the interaction of functionally associated proteins may result in a solubilization effect, we developed an *E. coli* system to co-express LmbA-like proteins with their upstream enzymes in the pathways, that is, LmbB1-like dioxygenases. Then, we examined the availability of soluble proteins using an *in vivo* activity assay in the presence of L-DOPA (**1**) prior to purification. The control systems included the recombinant *E. coli* strains, each of which expressed dioxygenase alone, and their abilities to transform **1** (colourless, $[M+H]^+$ *m/z*: calcd. 198.0761 for

$[C_9H_{12}NO_4]^+$, found 198.0756) into pyrroline **2** or **3** (bright yellow, $[M+H]^+$ *m/z*: calcd. 212.0553 for $[C_9H_{10}NO_5]^+$, found 212.0556) were confirmed by the characteristic colour change and high-performance liquid chromatography with mass spectrometric detection (HPLC-MS). The permutation of LmbA and its homologues with LmbB1 and its homologues (for example, Ant12 and SibV) revealed the active pair of Ant6 and Ant12 (Supplementary Fig. 3). Their combination in *E. coli* produced a new compound (**5**, $[M+H]^+$ *m/z*: calcd. 140.0706 for $[C_7H_{10}NO_2]^+$, found 140.0707) that shares its molecular weight with the expected product diene **4** (**5** was demonstrated *in vitro* to be a tautomer of **4**). Because the transformation of **2** or **3** occurred, the recombinant Ant6 protein should be active in *E. coli* and, thus, soluble under the conditions in which Ant12 was co-expressed.

We then purified Ant6 from the co-expressing *E. coli* system. Three related products with different molecular weights were obtained (Fig. 4a), consistent with the notion that Ant6 is a γ-GT-like Ntn-hydrolase fold protein that is capable of auto-catalytically processing its full-length (~67 kDa) precursor into the large (~47 kDa) and small (~20 kDa; this subunit appeared obscure, partially because of protein instability) subunits to construct a functional heterodimer (Supplementary Fig. 4). Precursor cleavage appeared to occur during the expression and purification process, and the full-length remains (~60% of Ant6-related protein products) could be unfolded/misfolded products, which were resistant to further conversion. The structure of the Ant6 precursor was predicted online using the I-TASSER approach, which revealed high similarity to the crystal structures of *Bl*-γ-GT-T399A (PDB: 4Y23) from *Bacillus licheniformis* and *Ec*-γ-GT-T391A (PDB: 2E0W) from *E. coli* K-12. Thus, both of these were selected for homology modelling to identify the proteolytic site of Ant6 (Supplementary Fig. 5). *Bl*-γ-GT-T399A and *Ec*-γ-GT-T391A are γ-GT precursor mimics[31,32], in which the catalytically important nucleophilic residue Thr is mutated to Ala, and they lack the self-processing activity for the hydrolysis that occurs between Thr and the immediately upstream residue Glu/Gln. Comparative analysis identified the corresponding residues D429 and T430 in the Ant6 precursor, which constitute a motif that is highly conserved in all LmbA-like proteins (Supplementary Fig. 6). This finding suggested that the hydrolytic cleavage of the Ant6 precursor proceeds between these two residues to produce the large (429-aa) and small (192-aa) subunits; these sizes are in good agreement with the experimental values described above. To validate the necessity/variability of D429 and T430 residues, we prepared Ant6 variants by single mutation of D429A, T430A, T430S and T430C and double mutation of D429A&T430A. Although Ant6-D429A, Ant6-T430A, Ant6-T430C and Ant6-D429A&T430A were observed solely in full-length forms and accordingly lost their self-processing abilities, the purification of Ant6-T430S revealed the ~47-kDa large and ~20-kDa small subunits that arise from the cleavage of the ~67-kDa precursor, albeit in lower yields (Fig. 4a). These findings demonstrate the specificity of Ant6's autoproteolytic activity in which the most similar residue Ser can partially substitute for T430 in the autoproteolytic process.

LmbA2991, the orthologue of LmbA in *S. lincolnensis*, was readily prepared in *E. coli*. As expected, the purification produced its (perhaps unfolded/misfolded) full-length (~67-kDa) precursor and the large (~46-kDa) and small (~21-kDa) subunits (Fig. 4b), and the quality achieved was much higher than that found using Ant6. Most LmbA2991 precursor products (~85% of the related protein products) were self-cleaved, as demonstrated by gel electrophoresis. Additionally, the cleaved products, particularly the small unit, appeared to be more stable than those produced from

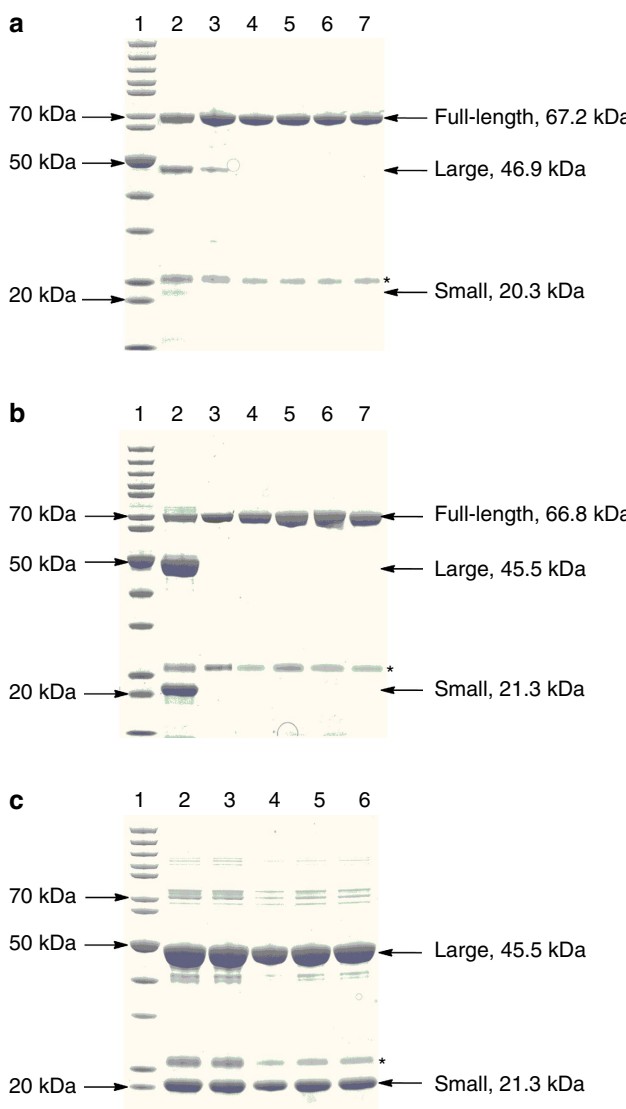

**Figure 4 | Analysis of autoproteolytic activity on SDS–PAGE gels.** The asterisk indicates an unknown impure protein associated with the purification of γ-GT homologues/variants. (**a**) Ant6. Lane 1, protein standard; Lane 2, Ant6 wild-type; Lane 3, Ant6-T430S; Lane 4, Ant6-T430C; Lane 5, Ant6-D429A&T430A; Lane 6, Ant6-T430A; Lane 7, Ant6-D429A. (**b**) LmbA2991. Lane 1, protein standard; Lane 2, LmbA2991 wild-type; Lane 3, LmbA2991-D420A; Lane 4, LmbA2991-T421A; Lane 5, LmbA2991-D420A&T421A; Lane 6, LmbA2991-T421C; Lane 7, LmbA2991 T421S. (**c**) Reconstituted LmbA2991 heterodimer (HD) large (L) and small (S). Lane 1, protein standard; Lane 2, heterodimer wild-type; Lane 3, LmbA2991 HD L-D420A&S; Lane 4, LmbA2991 HD L&S-T1A; Lane 5, LmbA2991 HD L&S-T1C; Lane 6, LmbA2991 HD L&S-T1S.

the Ant6 precursors. N-terminal sequencing of the purified products of LmbA2991 revealed two sequences: 'GSSHHHHHHS', which is shared by the full-length precursor and the 420-aa large subunit (consistent with the designed construct in which the target protein was produced in an N-terminally 6 His-tagged form), and 'TCHLDVVDRW', which is specific for the 197-aa small subunit (Supplementary Fig. 7), whose molecular weight was further established by high-resolution (HR)-MS (m/z: calcd. 21312.16, found 21312.63) (Supplementary Fig. 8). Similar to Ant6 precursor processing, the autoproteolysis of the LmbA2991 precursor occurs specifically between the conserved residues D420 and T421, leading

to the release of Thr, which resides at the N-terminus of the small subunit in the newly generated heterodimer. In contrast, both the single mutations of LmbA2991 (D420A, T421A, T421S and T421C) and the double mutation (D420A&T421A) completely abolished its autoproteolytic activity. Indeed, all enzyme variants were present solely in full-length precursor forms (Fig. 4b), indicating the stringency of LmbA2991 in self-processing, for which the key nucleophilic residue Thr is irreplaceable.

**Unusual Ntn-hydrolase activity for C-C bond cleavage.** Because pyrroline **2** or **3** is unstable and its purification and homogeneous chemical synthesis are difficult[19], we prepared this substrate in situ using LmbB1-catalysed oxidation reaction to examine the activities of Ant6 and LmbA2991 in vitro. LmbB1 was expressed and isolated from E. coli. Its dioxygenase activity was then reconstituted by incubation with $FeSO_4.7H_2O$, ascorbic acid and dithiothreitol[33], resulting in an active enzyme that was capable of rapidly converting L-DOPA into pyrroline **2** or **3** for the following tests (Fig. 5a).

The addition of the Ant6 catalyst, which was a mixture of the full-length and cleaved forms, into the above reaction mixture led to the nearly complete conversion of pyrroline **2** or **3** to **5** ($[M+H]^+$ m/z: calcd. 140.0706 for $[C_7H_{10}NO_2]^+$, found 140.0705) (Fig. 5a), the product also observed in the E. coli system in which both Ant6 and the dioxygenase Ant12 were co-expressed. This finding is consistent with the hypothesis that a diene product is yielded through the elimination of the terminal 2-C unit of the side chain. This conversion was scaled up, and subsequent spectral analysis suggested that **5** unlike **4**, which has previously been proposed to be an enamine resulting directly from the cleavage of **2** or **3**, is a tautomeric imine (Fig. 5b). To examine the tautomerization tendency of **4**, we synthesized a stable enamine mimic of **4** (**4'**, Fig. 5b), by protecting the carboxylate with a tert-butyl group and the amino group with a tert-butyloxy carbonyl group. Indeed, after deprotection, product analysis revealed **5** but no **4**, indicating that **4** is extremely unstable and tends to be converted to tautomer **5** by a diene shift. Meanwhile, the Ant6-containing reaction mixture was subjected to selective extraction and derivatization with N-methyl-N-(tert-butydimethylsilyl)-trifluoroacetamide (MTBSTFA) (Fig. 5b), leading to the observation of the derivative **8** by gas chromatography with MS detection (Supplementary Fig. 9). Therefore, the co-product is oxalic acid. This result explains the fate of the terminal 2-C unit of the side chain and confirmed that the cleavage of **2** or **3** proceeds through hydrolysis. The production of **5** was also observed in the reaction mixture in which Ant6 was replaced with an equal amount of LmbA2991 (Fig. 5a), confirming that LmbA2991 shares this unusual Ntn-hydrolase activity for C–C bond cleavage during the conversion of pyrroline **2** or **3**. However, the conversion efficiency of LmbA2991 was substantially decreased (~37% of that of Ant6) under the same reaction conditions, despite the catalyst mixture being of higher quality and stability and containing more cleaved (active) forms (as determined by the extent of self-activation).

**Catalytic similarities and differences of Ant6 and LmbA2991.** Taking advantage of the cascade assay system involving LmbB1 activity, we examined whether each mutant Ant6 or LmbA2991 catalyst functions in the conversion of pyrroline **2** or **3** to diene **5**. All full-length variants lacking autoproteolytic activity, including the derivatives from both Ant6 (that is, D429A, T430A, T430C and D429A&T430A) and from LmbA2991 (that is, D420A, T421A, T421S, T421C and D420A&T421A), were incapable of catalysing the hydrolysis of **2** or **3**; in contrast, the variant Ant6-T430S, which does possess autoproteolytic activity for

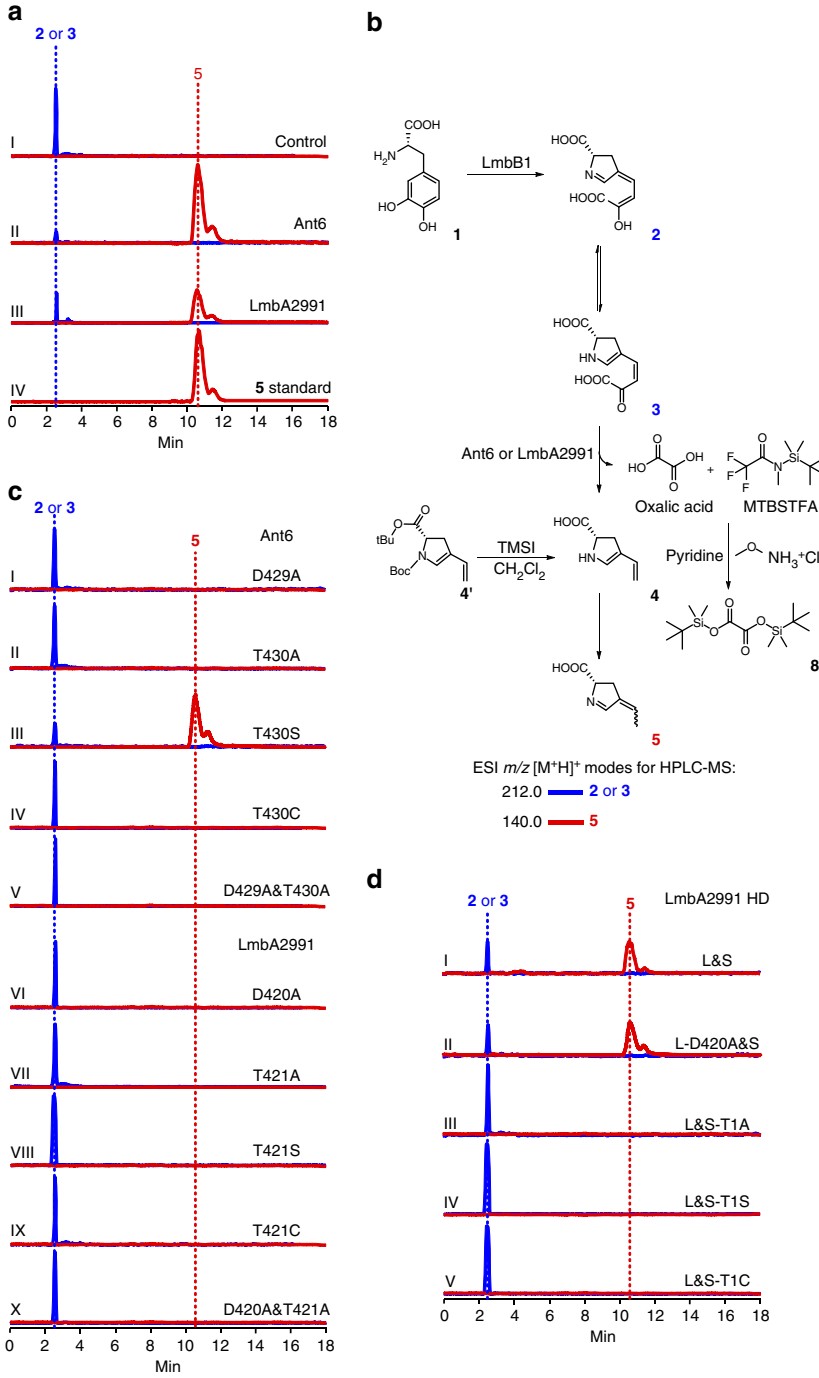

**Figure 5 | Examination of Ntn-hydrolase activity by HPLC-ESI-MS analysis.** Pyrroline substrate (**2** or **3**) was prepared *in situ* through LmbB1-mediated transformation of L-DOPA during the conversion to diene product (**5**) in the absence or presence of the catalysts. (**a**) Validation of Ntn-hydrolase activity of Ant6 and LmbA2991. I, negative control (without the catalyst Ant6 or LmbA2991); II, Ant6; III, LmbA2991 and IV, **5** standard. (**b**) Production of imine diene **5** through the tautomerization of enamine diene **4**, which results from either LmbB1 and Ant6 (or LmbA2991)-mediated cascade transformations (the derivatization of the co-product oxalic acid is indicated) or chemical deprotection of the synthesized mimic **4′**. (**c**) Evaluation of self-activation dependence for Ntn-hydrolase activity of Ant6 and LmbA2991. I, Ant6-D429A; II, Ant6-T430A; III, Ant6-T430S; IV, Ant6-T430C; V, Ant6-D429A&T430A; VI, LmbA2991-D420A; VII, LmbA2991-T421A; VIII, LmbA2991-T421S; IX, LmbA2991-T421C and X, LmbA2991-D420A&T421A. (**d**) Independent examination of Ntn-hydrolase activity of LmbA2991. I, LmbA2991 HD L&S; II, LmbA2991 HD L-D420A&S; III, LmbA2991 HD L&S-T1A; IV, LmbA2991 HD L&S-T1S and V, LmbA2991 HD L&S-T1C.

self-processing, was capable of this hydrolysis (Fig. 5c), albeit with activity lower than that of Ant6 wild type as judged by the yields of **5** and apparent reaction rates/efficiencies (Supplementary Fig. 10). These results demonstrate that the unusual Ntn-hydrolase activities of both Ant6 and LmbA2991 for C–C bond cleavage

completely depend on their autoproteolysis activities, because precursor cleavage is essentially a self-activation process.

To determine whether LmbA2991 shares the capacity for Ntn exchange with Ant6 (for example, between Thr and Ser/Cys) for the conversion of pyrroline **2** or **3** to diene **5**, we first

co-expressed the large and small subunits of LmbA2991 as two separated proteins and reconstituted the heterodimer through their interaction in *E. coli* (Fig. 4c and Supplementary Figs 4 and 11). This strategy by-passes the protein self-activation process because the LmbA2991-T421S/C precursor is resistant to autoproteolysis and cannot be converted to a functional heterodimer form; thus, we could evaluate the Ntn-hydrolase activities independently. The resulting LmbA2991 heterodimer formed an active complex in solution that exhibited hydrolytic activity towards **2** or **3** comparable to that of the catalyst prepared by LmbA2991 self-activation (Fig. 5d and Supplementary Fig. 10). We next evaluated the necessity/variability of D420 at the C-terminus of the large subunit, and in particular, T1 at the N-terminus of the small subunit; these two residues correspond to highly conserved D420 and T421 residues in the full-length LmbA2991 precursor. Subjecting the large subunit to D420A mutation had no effect on the conversion of **2** or **3** (Fig. 5d and Supplementary Fig. 10), thereby excluding D420 from playing a role in the C–C bond cleavage process; in contrast, subjecting the small unit to T1A, T1C or T1S mutation completely eliminated Ntn-hydrolase activity for the production of **5** (Fig. 5d). Clearly, unlike Ant6 catalysis, the role of the key residue Thr in LmbA2991 catalysis as an internal nucleophile for precursor splitting or as a newly released Ntn for carbon chain hydrolysis is absolutely irreplaceable.

## Discussion

In addition to γ-GTs, the Ntn-hydrolase fold proteins include glutamine 5-phosphoribosyl-1-pyrophosphate amidotransferase, penicillin acylase, proteasome subunit, glycosylasparaginase, isoaspartyl aminopeptidase and *N*-acylhomoserine lactone acylase that play critical roles in living organisms[2,34–39]. The members of this superfamily share a similar four-layered αββα structure and a common Ntn-based catalytic mechanism but do not have recognizable sequence homology. The maturation of each protein undergoes post-translational processing of a ribosomally synthesized precursor by autoproteolysis to release the characteristic and catalytically important Ntn, which is typically Thr, Ser or Cys. All Ntn-hydrolase fold proteins that have been identified to-date are amidohydrolases and catalyse various acyl transfer/hydrolysis reactions involving C–N bond cleavage by nucleophilic attack on the carbonyl carbon of the amide group of the substrate. To the best of our knowledge, enzymes with Ntn-hydrolase folds that catalyze C–C bond cleavage, as exemplified by Ant6 and LmbA2991, which were characterized in this study, have no precedent.

Ant6 and LmbA2991 are believed to share a self-activation process with the γ-GT-type Ntn-hydrolase fold proteins because they are phylogenetically related[1]. Precursor autoproteolysis occurs specifically between the highly conserved residues Glu/Asp and Thr, where the latter residue acts as an internal nucleophile to mediate the hydrolysis of the proceeding amide bond of the protein through the formation of a five-membered heterocycle intermediate and the exchange of an ester bond. This process usually results in a C-terminal Glu/Asp-containing large subunit and an *N*-terminal Thr-containing small subunit, which form an active heterodimer (Fig. 6a). The newly released *N*-terminal Thr residue, which may be relatively nucleophilic within the resulting catalytic framework[40], then functions at the active site to form an acyl-Thr enzyme intermediate for the subsequent acyl transfer/hydrolysis reaction (although other mechanisms, for example, the activation of $H_2O$ as a nucleophile for direct acyl hydrolysis, cannot be excluded at this time). However, unlike γ-GTs that are essentially amidohydrolases, Ant6 and LmbA2991 target the carbonyl group of the carbon side chain of the pyrroline intermediate and catalyse C–C bond cleavage

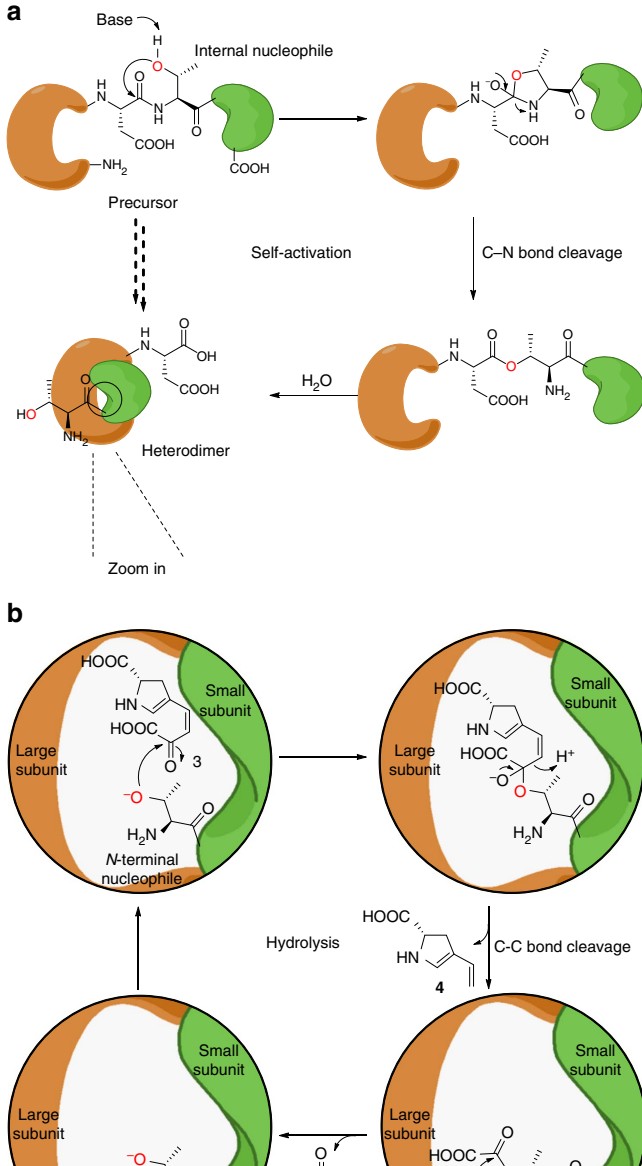

**Figure 6 | Proposed mechanisms of LmbA-like Ntn-hydrolase fold proteins.** (**a**) Self-activation through the autoproteolysis that proceeds between D429 and T430 of Ant6 and D420 and T421 of LmbA2991, respectively. (**b**) C–C bond cleavage by hydrolysis through the generation of oxalyl-Thr enzyme intermediate.

to yield the unstable enamine diene and oxalyl-Thr enzyme intermediate, the latter of which is subsequently hydrolysed to release oxalic acid as the co-product (Fig. 6b).

LmbA-like proteins represent the first γ-GT homologues characterized in actinomycetes. Their C–C bond cleavage activities highlight the versatile functions of Ntn-hydrolase fold proteins, which apparently have not been fully appreciated. Unlike LmbA, Ant6 and other related proteins that are specifically involved in the ALDP biosynthetic pathways (for example, Por11, SibY, TomL and HrmG), LmbA2991 is non-specific for ALDP formation and its intrinsic role in *S. lincolnensis*, the lincomycin A-producing strain, remains unclear. Phylogenetically, LmbA-like proteins fall into a clade close to but distinct from that of γ-GTs (Supplementary

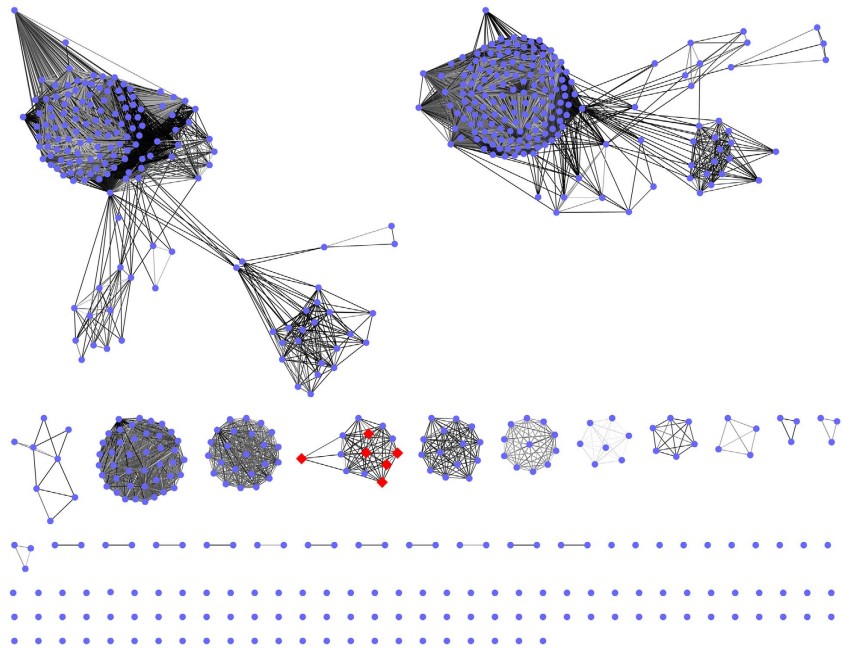

**Figure 7 | Sequence similarity networks of γ-GT homologues in Gram-positive actinobacteria.** To our knowledge, none of these γ-GT homologues (with an alignment score of 165) have been functionally characterized, with the exception of LmbA-like proteins (for example, LmbA, Ant6, Por11, SibY, TomL and HrmG, indicated by red diamonds) in this study. The sequence similarity networks are 95% representative node networks, that is, the sequences that share more than 95% identity are represented by a single node. The nodes that are not associated with Gram-positive actinobacteria were omitted manually.

Fig. 12). γ-GTs typically participate in GSH metabolism for γ-glutamyl transfer/hydrolysis in eukaryotes and Gram-negative bacteria. In contrast, in actinobacteria, including antibiotic-producing actinomycetes such as the producers of ALDP-containing PBDs, hormaomycins and lincomycin A, the dominant thiols that are functionally equivalent to GSH are mycothiol and EGT[3–5], neither of which contain γ-glutamyl in the structures. Therefore, it is not surprising that LmbA-like proteins functionally differ from GSH metabolism-associated γ-GTs. It is noteworthy that there are a number of homologues that have initially been annotated as γ-GTs in Gram-positive actinobacteria (Fig. 7). These homologues are functionally unassigned and could be different from LmbA-like proteins according to an extensive analysis of sequence comparison. Unveiling their roles in actinobacteria should be highly interesting, as unusual Ntn-hydrolase activities could be associated with new biochemical processes.

The study described herein represents a key step in elucidating the poorly understood biosynthetic pathways of various ALDP residues, as the shared reactions for the transformation L-tyrosine into the last common intermediate, pyrroline diene **4**, have been established. Following the actions of the oxidase pair, for example, LmbB1 and LmbB2, LmbA-like enzymes, including the ALDP pathway-specific Ntn-hydrolase fold proteins LmbA, Ant6, Por11, SibY, TomL and HrmG, are responsible for carbon side-chain tailoring. In the biosynthesis of lincomycin A, it is noteworthy that a non-specific orthologue, LmbA2991, functionally complements LmbA for the production of **4** in *S. lincolnensis*. The following conversion would proceed immediately because of the instability of intermediate **4**, either by hydrogenation or by *C*-methylation, and initiate a diversity-oriented process towards the biosynthesis of individual ALDP residues and associated natural products (Fig. 2b)[16].

In conclusion, we report a new type of γ-GT-like Ntn-hydrolase that catalyses C–C bond cleavage and provide mechanistic insights into the common pathway in the formation of various ALDP residues/associated natural products. These findings highlight the versatile functions of Ntn-hydrolase fold proteins, which are not limited to C–N bond cleavage as previously believed, expand the existing knowledge for biocatalysis in non-oxidative carbon-chain processing, and generate interest in numerous γ-GT homologues that are widely present in Gram-positive actinobacteria where their Ntn-hydrolase activities have not been well appreciated.

## Methods

**General.** General materials and methods are summarized in Supplementary Methods. Bacteria strains and plasmids used in this study are listed in Supplementary Table 1. Primers used in this study are summarized in Supplementary Table 2.

**DNA manipulation and sequencing.** DNA isolation and manipulation in *E. coli* and *Streptomyces* strains were performed according to standard protocols[41,42]. Polymerase chain reaction (PCR) amplification was conducted using an Applied Biosystems Veriti Thermal Cycler with either Taq DNA polymerase for routine genotype verification or PrimeSTAR HS DNA polymerase for high-fidelity amplification. Primer synthesis and DNA sequencing were performed at Shanghai Sangon Biotech Co. Ltd. (China) and Shanghai Majorbio Biotech Co. Ltd. (China), respectively.

**Gene inactivation and complementation.** For strain culture and fermentation, please see Supplementary Methods. The inactivation of *lmbA* (and *lmbA2991*) in *S. lincolnensis* was performed by in-frame deletion to exclude the polar effects on downstream gene expression. For *in trans* complementation, the target genes were under the control of *PermE\**, the constitutive promoter used to express the erythromycin-resistance gene in *Saccharopolyspora erythraea*. For details, please see Supplementary Methods.

**Sequence analysis.** Open reading frames were identified using the FramePlot 4.0beta program (http://nocardia.nih.go.jp/fp4/). The deduced proteins were compared with other known proteins in the databases using available BLAST methods (http://blast.ncbi.nlm.nih.gov/Blast.cgi). Amino acid sequence alignments were performed using the Strap program (http://www.bioinformatics.org/strap/). Homology modelling of the Ant6 structure was conducted using the I-TASSER online server (http://zhanglab.ccmb.med.umich.edu/I-TASSER/). Sequence

similarity networks were generated using the EFI-EST webtool (http://efi.igb.illi-nois.edu/efi-est/).

**Protein expression and purification.** LmbB1 was overexpressed in *E. coli* using the vector pET28a( + ) to give the *N*-terminally $6 \times$ His-tagged recombinant protein. The *N*-terminally $6 \times$ His-tagged Ant6 was co-expressed with the untagged, functionally associated upstream enzyme, Ant12, in *E. coli* using the vector pACYCDuet-1. The *N*-terminally $6 \times$ His-tagged LmbA2991 was co-expressed with the untagged LmbB1 in *E. coli* using the vector pACYCDuet-1. pACYCDuet-1 was also used to over-express the heterodimer LmbA2991 HD L&S, in which only the large subunit was tagged by $6 \times$ His at *N*-terminus. For details, please see Supplementary Methods.

**Site-specific mutations.** The plasmids containing site-directed mutations of Ant6, LmbA2991 and LmbA2991 HD L&S were generated using the primers listed in Supplementary Table 2 for PCR amplification with pLL2021, pLL2030 or pLL2034 as the template. Then, 1 μl of *Dpn*I enzyme was added to the PCR system to remove the template. After sequencing to validate the fidelity, the resulting mutant proteins were expressed in *E. coli* BL21 (DE3), purified to homogeneity, and concentrated according to the procedures described above for the native proteins.

**L-DOPA transformation in *E. coli*.** The recombinant plasmids for expressing LmbA + LmbB1, LmbB1 alone, Ant6 + Ant12, Ant12 alone, SibY + SibV, and SibV alone were transferred into *E. coli* BL21 (DE3), respectively, for expression at 16 °C and 220 rpm for 36 h. In each case, isopropyl-β-D-thiogalactopyranoside was added (to a final concentration of 100 μM) for protein-expression induction when the $OD_{600}$ value reached 0.6. The cells were cultured at 28 °C for 28 h after the addition of 2 mM L-DOPA (final concentration). The supernatant was collected, concentrated and analysed by HPLC-electrospray ionization-MS using a Phenomenex column (Luna 5 μ C18(2) 100A, 4.60 × 250 mm, 5 micron, phenomenex Inc., USA) with a gradient elution of solvent A ($H_2O$ containing 20 mM $NH_4OAc$) and solvent B ($CH_3OH$) at a flow rate of 1 ml min$^{-1}$ and ultraviolet (UV) detection at 240 nm using a 30-min gradient program: $t = 0$ min, 5% B; $t = 5$ min, 5% B; $t = 15$ min, 15% B; $t = 25$ min, 15% B; $t = 26$ min, 5% B and $t = 30$ min, 5% B.

**In vitro enzymatic assay.** The assays (total volume, 100 μl) were performed at 30 °C for 3 h in 100-mM $K_2HPO_4$ buffer (pH 8.0) containing approximately 500-μM L-DOPA in the presence of 10-μM LmbB1 and 10-μM Ant6 (LmbA2991 or LmbA2991 HD L&S) or mutant enzymes. LmbB1 was added last to initiate the reaction. The assays were quenched by adding an equal volume of methanol. For the examination of pyrroline intermediate **2** or **3** and imine diene **5**, after centrifugation (5 min at 13,000*g*), the supernatant was subjected to HPLC-electrospray ionization-MS analysis following the method described above. For the examination of the co-product oxalic acid, please see Supplementary Methods. For quantitative comparison in C–C bond cleavage activity, the large subunit of each γ-GT homologue, which is stable than the small subunit in the heterodimer, served as an indicator to quantify the active form of γ-GT homologue to a same concentration. Specifically, for quantitative comparison of the activities of Ant6 wild-type and variant Ant6-T430S, The assays were performed at 30 °C for 4 h using 10-μM Ant6 wild type and 30-μM Ant6-T430S, respectively; and for quantitative comparison of the activities of LmbA2991 wild type, the reconstituted heterodimer LmbA2991 HD L&S and its variant LmbA2991 HD L-D420A&S. The assays were performed at 30 °C for 4 h using 11.5-μM LmbA2991 wild type, 10-μM LmbA2991 HD L&S and 10-μM LmbA2991 HD L-D420A&S, respectively. All assays were performed in triplicate.

**Chemical synthesis.** The chemicals including **4′** and **5** were synthesized according to the methods previously described[43–45]. For details, please see Supplementary Methods.

**Data availability.** The sequence of the gene *lmbA2991* is deposited in GenBank with the NCBI accession numbers KY987495. Data supporting the findings of this study are available within the article and its supplementary information files and from the corresponding author on reasonable request.

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

## Acknowledgements

This work was supported in part by grants from NSFC (21520102004, 31430005, 81302674 and 21472231), CAS (QYZDJ-SSW-SLH037 and XDB20020200), STCSM (14JC1407700 and 15JC1400400), K.C. Wang Education Foundation and Chang-Jiang Scholars Program of China.

## Author contributions

G.Z., Q.Z. and Q.-l.Z. performed the *in vivo* genetic investigations. G.Z. conducted the *in vitro* biochemical studies. G.Z. and Q.Z. performed the sequence analysis. G.Z. conducted the chemical synthesis and purification. G.Z. and W.L. analysed the data and wrote the manuscript. W.L. directed the research.

## Additional information

**Competing interests:** The authors declare no competing financial interests.

