## [Peer Review File · Nature Communications]

Reviewer #1 (Remarks to the Author):

The article by Zhong and coworkers reports on an enzyme involved in the biosynthesis of 4-alkyl-L-(dehydro)proline (ALDP) derivatives. These unusual amino acids are found in a few different classes of peptidic natural products, many of which bear attractive pharmacological properties, and the nature by which they are constructed have, until now, remained elusive. In this report, the authors demonstrate that enzyme belonging to the Ntn-hydrolase family, bearing an N-terminal residue as a catalytic nucleophile, adapt the active site scaffold to mediate the cleavage of a C-C bond. This is a particularly remarkable finding given that all other Ntn-hydrolases identified to date facilitate cleavage of an amide, or C-N, bond via a covalent intermediate. The implications of this work is far reaching: as the authors highlight, there is a large number of Ntn-hydrolase-like proteins encoded within bacterial genomes of unknown function. Knowledge that this enzyme family can effect more diverse chemistry certainly alters the perspective we now have and it is likely that appreciation of new functions will soon emerge. The manuscript is very well written and easy to follow, and the experiments appear to be conducted according to the standards of the field. This work is very appropriate for publication in Nature Communications, and will be of great interest to the broad readership of the journal. I have only a few concerns, mostly minor, which are detailed below.

If at all possible, some type of kinetic/activity data should be provided for all enzymes and constructs included in this work. Ideally, there needs to be a more quantitative comparison of wild-type activities to that of site-directed mutants, the reconstituted heterodimer, and site-directed mutants of the reconstituted heterodimer.

Additionally, though less important for this work, it could be worthwhile to include experimental data that would address heterodimer formation (both native and reconstituted) beyond a relatively qualitative activity assay. Perhaps the inclusion of gel electrophoresis or another method that quantifies the strength of the protein-protein interaction (i.e. a Kd).

Can the authors elaborate on a proposed mechanism for the C-C cleavage beyond what is shown in the figure and currently discussed? At a first glance, this looks to be a challenging elimination, as the pKa of the sp² carbon would likely be quite high.

Centrifugation information (page 19, line 14; and in supporting information experimental) should include x g in addition to (or in place of) rpm. The x g can vary between instruments from identical rpm and thus provides a more reliable way for someone to reproduce the procedure.

Other comments:

Remarkably decent English for a W. Liu paper; must have a fluent PDF or grad student.

Page 7: Statement about PPL supply; comma is needed after maturation.

Figure 4: When referring to samples in SDS-PAGE analysis (would denature dimers), the use of the term "native" and may be potentially confused with Native PAGE analysis (would keep dimers folded or "native" and not lead to strand separation via PAGE). Consider rephrasing...perhaps to "wild-type"? It is also unclear why SDS-PAGE of the reconstituted heterodimer (panel c) is necessary to include in the main text...It certainly demonstrates the purity of the proteins, but doesn't provide information on successful reconstitution of the dimer, like a native PAGE or Gel electrophoresis or other biophysical technique would.

Page 9, line 19: gel electrophoresis rather than gelelectrophoresis

Page 11, lines 6-7: In the sentence starting with "To examine its tautomerization tendency..." it is unclear what "its" is referring to, given that prior sentence is referring to 4 different analogs.

Page 13, line 20: amidohydrolase rather than aminohydrolase

Page 15, line 6: hormaomycins rather than homaomycins?

Page 15, line 11: unassigned rather an unsigned?

Page 18, line 12: Dpn should be italicized

Page 19, line 3: Phenomenex should be capitalized

Page 20, line 17: Streptomyces should be italicized

Reviewer #2 (Remarks to the Author):

In this manuscript by Zhang, they outlined their efforts on characterized a unusual sub-class of GTS. The discovery of C-C bond cleavage activity is definitely a very important discovery and this paper is worth publishing on Nature Communication.

The work here is very though, including genetics, protein expression, and detailed biochemical characterizations. The evidence presented are all consistent with the proposed activity.

Please incorporate some of my minor concerns:

for the proposed mechanism in Fig.6, the authors proposed Thr as a nucleophile and the involvement of a covalent intermediate, which is a reasonable hypothesis. Please also consider the option of having H₂O as the nucleophile, activated by the same Thr residue.

Overall, this is an extremely well-written manuscript. Easy to follow and with extensive data to support the proposed unusual activity. I support the publication of this manuscript with the suggested minor modification.

Reviewer #3 (Remarks to the Author):

The authors describe the characterization of γ -glutamyltranspeptidases (γ -GT) homologues found in actinomycetes that play a vital role in biosynthesis of 4-Alkyl-L-(dehydro)proline (ALDP) residues in many important secondary metabolites. Even though the γ -GTs in eukaryotes and Gram-negative bacteria are well known to be involved in thiol glutathione metabolism, the functions of γ -GTs homologues that are widely present in Gram-positive actinobacteria remain largely unknown. Here, the authors provide compelling evidence to demonstrate that some LmbA-like γ -GTs homologues are capable of cleaving C-C bond in pyrroline, a common intermediate in ALDP biosynthetic pathways, to form diene and oxalic acid as a co-product. The LmbA-like enzymes are proven to be important for ALDP containing natural product biosynthesis via gene knockout leading to decreased yield or total loss of lincomycin A production. The LmbA and its counterparts from other actinomycetes are able to form diene intermediate in vivo (when heterologously expressed in *E. coli* system) and in vitro. Similar to known γ -GTs, LmbA-like proteins undergo self-activation via posttranslational autocatalytic processing, and a proposed mechanism of hydrolytic C-C bond cleavage through similar nucleophilic attack on the carbonyl group of the carbon side chain of pyrroline is also provided and partially supported by site-directed mutagenesis. The experiments are clearly demonstrated and results are solid and interesting. Here are some minor comments:

- Figure 2c or S5. The readers may value identity/similarity information among encoding proteins that are conserved in ALDP-containing metabolites, at least LmbA and its counterparts.
- Page 5, line 11. A citation is suggested to be added.

- Page 8, line 1. Please revise the confusing sentence “various tests revealed the presence of the pair of Ant6 and the dioxygenase Ant12”. Did any other pairs (e.g. LmbA/LmbB1 pair) show similar activity? Why Ant12 helped solubility of Ant6? Any specific protein-protein interaction between them? Did other dioxygenase help solubility of Ant6? Please provide more results and additional comments in the discussion.
- Page 9, line 3 and SI fig. 5. DT motif is indeed highly conserved among LmbA-like proteins, but the autocatalytic environment (interaction between Asp/Thr and remote Arg/Glu, as discussed in details in Ref. 32) still needs to be considered.
- Figure 4. What is the protein band just above 20 kDa that seems to be in all gels?
- Figure 5d. Are both large and small subunits required for the C-C bond cleavage?

Response to the Reviewers' Comments

We have revised the manuscript according to the comments from the reviewers. The specific responses to the reviewers' points (shown in *italic*) are summarized below.

1. *If at all possible, some type of kinetic/activity data should be provided for all enzymes and constructs included in this work. Ideally, there needs to be a more quantitative comparison of wild-type activities to that of site-directed mutants, the reconstituted heterodimer, and site-directed mutants of the reconstituted heterodimer (reviewer 1#).*

We appreciate this constructive suggestion. All γ -Glutamyltranspeptidase (γ -GT) homologs/variants in this study that exhibit Ntn-hydrolase activity were chosen and grouped into two sets for quantitative comparison in C-C bond cleavage activity, i.e., Ant6 wild-type vs variant Ant6-T430S and LmbA2991 wild-type vs the reconstituted heterodimer LmbA2991 HD L&S and its variant LmbA2991 HD L-D420A&S. Since substrate **2** (pyrroline imine) or **3** (pyrroline enamine) is unstable and was prepared *in situ* (by transforming L-DOPA) using LmbB1-catalyzed oxygenation reaction, the activities of γ -GT homologs were analyzed according to time courses in the formation of **5** (a tautomer of product **4**) over a 4-h period. In this comparison (as judged by the yields of **5** and apparent reaction rates/efficiencies), the large subunit of each γ -GT homolog, which is stable than the small subunit in the heterodimer, served as an indicator to quantify the active form of γ -GT homolog to a same concentration (i.e., 10- μ M Ant6 wild-type equal to 30- μ M Ant6-T430S and 11.5- μ M LmbA2991 wild-type to 10- μ M LmbA2991 HD L&S or 10- μ M

LmbA2991 HD L-D420A&S (Please see the 2nd paragraph of Page 19 in **Materials and Methods** of the revised main text). Consequently, Ant6-T430S exhibited Ntn-hydrolase activity lower than that of Ant6 wild-type (**left, a**), in contrast to LmbA2991 wild-type, LmbA2991 HD L&S, LmbA2991 HD L-D420A&S, which displayed comparable activity in C-C bond cleavage (**right, b**). We have accordingly added these data into the revised main text (Please see the 1st paragraph of Page 12 and the 1st paragraph of Page 13 in the revised main text) and Supplementary Fig. 10 in **Supplementary Information** (Please see Page 28).

2. *Additionally, though less important for this work, it could be worthwhile to include experimental data that would address heterodimer formation (both native and reconstituted) beyond a relatively qualitative activity assay. Perhaps the inclusion of gel electrophoresis or another method that quantifies the strength of the protein-protein interaction (i.e. a Kd) (reviewer 1#)?*

(a) Non-denaturing gradient PAGE analysis (conc. 4% to 16%). Lane 1, Ant6 wild-type; Lane 2, Ant6-T430S; Lane 3, LmbA2991 wild-type; Lane 4, LmbA2991 HD L&S; Lane 5, LmbA2991 HD L-D420A&S. **(b)** Denaturing SDS-PAGE analysis (conc. 10%). Lane 1, protein standard; Lane 2, Ant6 wild-type; Lane 3, Ant6-T430S; Lane 4, LmbA2991 wild-type; Lane 5, LmbA2991 HD L&S; Lane 6, LmbA2991 HD L-D420A&S.

To further support that the active γ -GT homologs/variants (i.e., Ant6 wild-type and Ant6-T430S, and LmbA2991 wild-type, the reconstituted heterodimer LmbA2991 HD L&S and its variant

LmbA2991 HD L-D420A&S) function in heterodimeric forms, we performed an analysis of these proteins on both non-denaturing gradient PAGE (conc. 4% to 16%) (**Left, a**) and denaturing SDS-PAGE (conc. 10%) (**Right, b**). These γ -GT homologs/variants appeared exclusively to be one product for each under non-denaturing conditions, whereas under denaturing conditions, the full-length form (for Ant6 wild-type, Ant6-T430S and LmbA2991 wild-type) and/or the cleaved form (as shown by two subunits that differ in MW for all test proteins) were observed. In particular, both the heterodimers LmbA2991 HD L&S and LmbA2991 HD L-D420A&S were reconstituted only under conditions where the large and small subunits were co-expressed in *E. coli* (please see Protein expression and purification in **Materials and Methods** of the main text and Supplementary Methods in **Supplementary Information**). Although only the large subunit was tagged by 6 x His at N-terminus during each reconstitution, the purification on a Ni-NTA column resulted in both the large and 6 x His-**untagged** small subunits. In contrast, soluble protein preparation failed when separately overexpressing the large or small subunit in *E. coli*.

Further, we conducted an analysis of the reconstituted heterodimer LmbA2991 HD L&S on FPLC, where a sole peak was observed (**top, a**). This fraction was collected and then subjected to

SDS-PAGE analysis, which did reveal both the large and small subunits (**bottom, b**), with a ratio of approximately 1:1. These results strongly supported that the both subunits works together to form a functional heterodimer.

We agree with the reviewer that quantitative analysis of the strength of the protein-protein interaction (i.e. a Kd) is helpful; however, the fact that the large or small subunit could not be prepared separately hampers attempts in measurement (e.g., using isothermal titration calorimetry). Since the above findings supported that the interaction of the large and small subunits does exist in a functional heterodimer of each γ -GT homolog/variant, we have incorporated these new data into the revised manuscript (the 2nd paragraph of Page 8 and the 2nd paragraph of Page 12). Please see Supplementary Figs. 4 and 11 in **Supplementary Information**.

3. *Can the authors elaborate on a proposed mechanism for the C-C cleavage beyond what is shown in the figure and currently discussed? At a first glance, this looks to be a challenging elimination, as the pKa of the sp² carbon would likely be quite high (reviewer 1#).*

We thank the reviewer for this suggestion. We are hesitating to provide a more detailed mechanism for C-C bond cleavage as shown in Fig. 6 before we solve the structure of LmbA2991 (the crystallization of LmbA2991 is on-going). The structural insights into its active site would be helpful in understanding how the catalytic environment impacts on substrate **3**, e.g., through the interaction with its pyrroline-containing, double bond-conjugated system, to lower the pKa of the sp² carbon.

4. *Centrifugation information (page 19, line 14; and in supporting information experimental) should include x g in addition to (or in place of) rpm. The x g can vary between instruments from identical rpm and thus provides a more reliable way for someone to reproduce the procedure (reviewer 1#).*

We have revised centrifugation information as the reviewer suggested.

5. *Page 7: Statement about PPL supply; comma is needed after maturation (reviewer 1#).*

We have added comma as the reviewer suggested.

6. *Figure 4: When referring to samples in SDS-PAGE analysis (would denature dimers), the use of the term “native” and may be potentially confused with Native PAGE analysis (would keep dimers folded or “native” and not lead to strand separation via PAGE). Consider rephrasing...perhaps to “wild-type”? It is also unclear why SDS-PAGE of the reconstituted heterodimer (panel c) is necessary to include in the main text...It certainly demonstrates the purity of the proteins, but doesn’t provide information on successful reconstitution of the dimer, like a native PAGE or Gel electrophoresis or other biophysical technique would (reviewer 1#).*

We have changed the word “**native**” to “**wild-type**” for protein description throughout the main text and **Supplementary Information** as the reviewer suggested. Please see the revised manuscript. If we have space, we would prefer to include panel c of Fig. 4 for SDS-PAGE analysis of the reconstituted heterodimer LmbA2991 HD L&S and its variants. In addition to the purity of the proteins, this panel shows a major difference from panel b: the full-length precursors of γ -GT homologs/variants disappeared. Coupling with the newly incorporated data (as the reviewer suggested) concerning selective analysis of active proteins on native (non-denaturing) gradient PAGE and particularly FPLC analysis of the heterodimeric nature of LmbA2991 HD L&S (please refer to the above **response to Point 2**), panel c now becomes informative and would provide information on successful reconstitution of related dimers.

7. *Page 11, lines 6-7: In the sentence starting with “To examine its tautomerization tendency...” it is unclear what “its” is referring to, given that prior sentence is referring to 4 different analogs (reviewer 1#)*

We have changed this sentence to “To examine the tautomerization tendency of **4**”.

8. Page 9, line 19: *gel electrophoresis rather than gelelectrophoresis*; Page 13, line 20: *amidohydrolase rather than aminohydrolase*; Page 15, line 6: *hormaomycins rather than homaomycins*; Page 15, line 11: *unassigned rather an unsigned*; Page 18, line 12: *Dpn should be italicized*; Page 19, line 3: *Phenomenex should be capitalized*; Page 20, line 17: *Streptomyces should be italicized (reviewer 1#)*.

Many thanks for the help in proof-reading. We have corrected these errors indicated by the reviewer. Please see the revised manuscript.

9. For the proposed mechanism in Fig. 6, the authors proposed Thr as a nucleophile and the involvement of a covalent intermediate, which is a reasonable hypothesis. Please also consider the option of having H₂O as the nucleophile, activated by the same Thr residue (reviewer 2#).

We appreciate this constructive suggestion. Accordingly, we have revised the abstract and related discussion in the main text: “**likely** through the generation of an oxalyl-Thr enzyme intermediate” in Page 2 and “although other mechanisms, e.g., the activation of H₂O as a nucleophile for direct acyl hydrolysis, cannot be excluded at this time” in Page 14. The prediction that Ant6/LmbA2991 catalysis involves a covalent intermediate, as shown in Fig. 6, is consistent with the well-established common mechanism for all characterized Ntn-hydrolase fold proteins.

10. Figure 2c or S5. The readers may value identity/similarity information among encoding proteins that are conserved in ALDP-containing metabolites, at least LmbA and its counterparts (reviewer 3#).

We thank the reviewer for this suggestion. Accordingly, we have added information into the legend of Fig. 2: “For sequence identities of deduced proteins, LmbA is homologous to Ant6 (72.3 %), Por11 (72.9 %), SibY (58.0 %), TomL (50.0 %) and HrmG (64.6 %); LmbY is homologous to Ant14 (49.8 %), Por15 (48.2 %), SibT (53.5 %), TomJ (50.2 %) and HrmD (46.3 %), LmbB1 is homologous to Ant12 (40.8 %), Por13 (44.6 %), SibV (54.4 %), TomH

(52.8 %) and HrmF (46.2 %); and LmbB2 is homologous to Ant13 (31.2 %), Por14 (30.9 %), SibU (38.7 %), TomI (37.3 %) and HrmE (42.9 %)”. Please see revised Fig 2 in the main text. The identity of LmbA2992 (48.4%) to LmbA is indicated in the main text (please see the 3rd paragraph in Page 6).

11. Page 5, line 11. A citation is suggested to be added (reviewer 3#).

We have added a citation as the reviewer suggested.

12. Page 8, line 1. Please revise the confusing sentence “various tests revealed the presence of the pair of Ant6 and the dioxygenase Ant12”. Did any other pairs (e.g. LmbA/LmbB1 pair) show similar activity? Why Ant12 helped solubility of Ant6? Any specific protein-protein interaction between them? Did other dioxygenase help solubility of Ant6? Please provide more results and additional comments in the discussion (reviewer 3#).

As the reviewer suggested, we have changed this sentence to “The permutation of LmbA and its homologs with LmbB1 and its homologs (e.g., Ant12 and SibV) revealed the active pair of Ant6 and Ant12”. Please see 1st paragraph of Page 8 in the revised manuscript. Except the pair of Ant6 and Ant12, other test combinations (including the pair of LmbA and LmbB1) failed to produce **5** (a tautomer of product **4**). As mentioned in the main text, we did have long been hampered by the failure in the preparation of soluble LmbA-like proteins. Among our numerous attempts, the co-expression with upstream LmbB1-like dioxygenases in *E. coli* was such a way that facilitates a rapid in vivo assay of **5** production to judge whether soluble products of LmbA-like protein are available. We were lucky enough using this way to identify the pair of Ant6 and Ant12 and thus were able to purify Ant6 from the co-expressing *E. coli* system for further in vitro activity assays. We haven’t followed the question of why Ant12 (rather than other dioxygenases, e.g., LmbB1 or SibV) helped the solubility of Ant6 (because this could take time and is not the focus of this manuscript); instead, we took advantage of this result to obtain the recombinant protein and paid our attention in this study to a subject of much concern of whether Ant6 shares mechanistic similarity to γ -GTs and exhibits autoproteolytic activity and subsequent Ntn-hydrolyse activity.

13. Page 9, line 3 and SI fig. 5. DT motif is indeed highly conserved among LmbA-like proteins, but the autocatalytic environment (interaction between Asp/Thr and remote Arg/Glu, as discussed in details in Ref. 32) still needs to be considered (reviewer 3#).

We agree with the reviewer that the catalytic environment is of importance to each γ -GT homolog/variant activity. In this study, we analyzed the necessity of the highly conserved residue Asp and Thr, which can be readily revealed by sequence alignment or homology modeling, to both autoproteolytic activity of the precursor and Ntn-hydrolyse activity of the mature heterodimer; however, we are hesitating to expand the discussion to other residues that are potentially necessary for autoproteolysis and C-C bond cleavage in the absence of solid structural information. The crystallization of LmbA2991 is in progress, and structural insights into its active site would facilitate the identification of these residues that are essential to LmbA2991 activity through the interactions with conserved residues Asp and Thr or with the pyrroline-containing, double bond-conjugated system of substrate **3**.

14. Figure 4. What is the protein band just above 20 kDa that seems to be in all gels (reviewer 3#)?

This band indicates an unknown impure protein associated with the purification of γ -GT homologs/variants. We have added a note to the legend of Fig. 4. Please see the revised manuscript.

15. Figure 5d. Are both large and small subunits required for the C-C bond cleavage (reviewer 3#)?

Yes. The large and small subunits work together to constitute an active heterodimer of each γ -GT homolog/variant for C-C bond cleavage. In fact, soluble protein preparation failed when separately overexpressing the large or small subunit in *E. coli*. For evidence that each γ -GT homolog/variant functions in a heterodimeric form, please refer to the response to **Point 2**, Fig. 4 in the main text

and Supplementary Figs. 4 and 11 in **Supplementary Information**.

Reviewer #1 (Remarks to the Author):

This referee is satisfied with the alterations and additions to this much-improved manuscript.

Reviewer #3 (Remarks to the Author):

The revised manuscript has been improved and the authors have addressed most of the concerns and comments raised by reviewers. It is suitable for publication with no further changes.